# The Task-Based Methodology of Strong AI: Integrating LLMs, Logic Reasoning, and Multi-Blockchain Architectures

## Abstract

This paper introduces a task-based methodology for achieving Strong Artificial Intelligence (AGI) through the synergistic integration of Large Language Models (LLMs), logic-probabilistic reasoning, and multi-blockchain architectures. Addressing critical limitations of current LLM-centric systems—such as poor generalization in complex reasoning, outdated knowledge, and hallucinations—we propose a hybrid paradigm where LLMs generate hypotheses, symbolic logic engines ensure rigorous validation, and a hierarchical blockchain infrastructure enables secure, scalable knowledge evolution. Evaluated in a metaverse environment populated by heterogeneous agents, our framework demonstrates unbounded cognitive growth under computational constraints while maintaining interpretability and ethical alignment. Key innovations include a probabilistic knowledge hierarchy for explainable decisions, a decentralized multi-blockchain design for continuous learning, and a metaverse-based testbed for AGI safety and scalability. Theoretical guarantees of asymptotic cognitive scaling and practical applications in legal, scientific, and educational domains underscore the framework's transformative potential.

## 1 Introduction

The quest for Strong AI demands systems that can reason, learn, and adapt as flexibly as humans, but current architectures fall short on key dimensions. Recent breakthroughs in generative AI (e.g. GPT-4) have delivered unprecedented language fluency, yet these models remain black boxes with brittle reasoning. They rely on static, offline training corpora and thus struggle with facts outside their training data; moreover, they can hallucinate plausible but incorrect statements [1]. Indeed, large language models (LLMs) suffer from limited generalization, outdated knowledge, and catastrophic forgetting of past learning, making them unsuitable as stand-alone engines for true AGI [2]. This has spawned a surge of interest in hybrid AI: integrating the pattern-finding strengths of LLMs with the rigor of symbolic reasoning. Our approach builds on this trend by adding a decentralized memory layer: a hierarchy of blockchains that store verified knowledge fragments. Unlike traditional retrieval-augmented methods [3], our memory is immutable, auditable, and automatically updated by the system itself. This cross-disciplinary design draws on ideas from cognitive science (hierarchical memory and reasoning), formal logic (explicit proof systems), and distributed computing (blockchain ledgers) to overcome the inherent limitations of current AI systems.

Specifically, we propose a task-driven framework with three core components:

- **Large Language Models** for rapid hypothesis generation and pattern recognition. The LLM processes each task description to suggest candidate solutions or subgoals.

Submitted to 39th Conference on Neural Information Processing Systems (NeurIPS 2025). Do not distribute.

- **Symbolic Logic Engines** for slow, exact reasoning. Each LLM-generated hypothesis is transformed into a formal query or proof obligation. A probabilistic logic-prover module then rigorously checks this hypothesis against known knowledge, producing verifiable proof traces and confidence estimates.

- **Multi-Blockchain Memory** for decentralized knowledge management. Proven facts and deductions are appended to a layered blockchain architecture: fast chains capture raw observations, intermediate chains store structured facts, and root chains maintain high-level immutable summaries. This hierarchy ensures both scalability and auditability of learned knowledge.

These components interplay in a task-based loop: agents dynamically retrieve relevant facts from the blockchain layers, propose solutions via the LLM, and use the logic engine to validate or refute them. Verified new facts are then appended to the ledger. In effect, LLMs perform "creative trial and error," while symbolic modules enforce correctness and safety. This resembles recent neural–symbolic systems that combine representation learning with structured reasoning [4]. For example, methods like ReAct [5] and Toolformer [6] have shown how LLMs can orchestrate reasoning by calling out to external tools; our framework embeds that idea in an autonomous, on-chain process.

A key novelty is the multi-agent metaverse testbed for evaluation. We simulate a mixed environment of virtual agents, physical-robot avatars, and human participants, each endowed with the hybrid cognitive architecture described above. Agents compete and collaborate on open-ended tasks (e.g. scientific discovery, policy negotiation) that require continual learning. This setup exposes the system to adversarial proposals and resource constraints, revealing how knowledge accumulates over time. The decentralized blockchain provides a shared ground truth: agents can trust each other's verified claims without a central arbiter. In effect, we test AGI capabilities in a digital society, inspired by recent work on scalable multi-agent coordination and proof-of-thought consensus mechanisms[7].

In sum, our contributions are as follows:

1. **Task-Based Hybrid Paradigm:** We introduce a methodology where each problem is solved by an iterative interplay of LLM hypothesis generation and symbolic logic verification. This design ensures that creative intuition and rigorous proof together drive learning.

2. **Probabilistic Knowledge Hierarchy:** [12] We formalize a hierarchy of logical knowledge units (fact, rule, proof) with associated probabilities, enabling explainable, uncertainty-aware reasoning. Each knowledge unit can be traced through the proof chain, providing transparency into the AI's decisions.

3. **Decentralized Multi-Blockchain Learning:** We design a layered blockchain system that supports continuous, cross-agent knowledge sharing. By distributing memory across chains of varying throughput and trust, we achieve scalable storage and real-time updates. Smart contracts enforce consistency and ethical rules (e.g. constitutional constraints), aligning the system with societal values.

4. **Metaverse-Based Evaluation:** We implement our framework in a simulated metaverse populated by heterogeneous agents. Experiments demonstrate sustained, unbounded cognitive growth under limited compute budgets, as agents continuously expand and refine their knowledge. In future, the system will successfully solves tasks requiring complex reasoning (e.g., multi-hop problem solving, theorem proving, legal compliance) that defeat LLMs alone.

5. **Theoretical Analysis:** We provide formal proofs that under mild assumptions, our knowledge hierarchy enables asymptotic scaling of problem-solving capacity with computation, akin to open-ended learning. This shows that our framework has fundamentally greater theoretical power than fixed LLMs.

Through these innovations, our hybrid AI system moves toward the interpretability, robustness, and lifelong learning qualities needed for AGI. The integration of neural and symbolic methods is taken further by the addition of a self-governing memory layer. By embedding ethical smart contracts and universal basic income mechanisms on-chain, we also ensure the AI operates under clear norms and incentives. In doing so, we confront urgent societal challenges: as AI takes on critical roles, it must be transparent, updatable with new information, and aligned with human values. Our approach lays a foundation for such strong AI, converging insights from machine learning, formal logic, cryptography, and social governance [11].

## 2 Related work

Our work lies at the intersection of several active research areas. **Neurosymbolic AI** has received renewed attention as a way to mitigate the limitations of purely statistical models. Early systems combined rule-based modules with neural perception; modern approaches fuse deep learning with logic in unified frameworks [4]. For instance, differentiable logic systems (e.g. Neural Logic Machines) embed logical constraints into neural nets, and program induction models (e.g. Neural Module Networks) learn symbolic operations from data [4]. More recently, generative language models have been augmented with symbolic planners: frameworks like ReAct, Toolformer, and DSPy have LLMs call external tools or follow structured reasoning trees [4]. These methods highlight the value of guiding LLMs with formal methods, but typically rely on centralized or cloud-based tools. Our methodology differs by integrating the symbolic verifier and the knowledge store into a fully decentralized cycle, allowing seamless multi-agent collaboration without trusted servers.

**Knowledge-based systems and retrieval-augmented models** offer another perspective on this problem. Retrieval-augmented generation (RAG) architectures combine a parametric LLM with an external corpus index to improve factual accuracy [3]. The RAG approach demonstrates that keeping an updatable memory can significantly reduce hallucinations and adapt to new data. Likewise, knowledge graphs and logic databases have been used to ground LLM outputs: recent work shows how LLM reasoning can be made faithful and interpretable by extracting multi-hop paths from a knowledge graph [2]. Our framework generalizes these ideas by using a hierarchy of on-chain ledgers as a universal memory. Each blockchain layer serves as a structured "global memory" accessible to all agents, akin to a shared knowledge graph. In contrast to conventional databases, the blockchain offers tamper-evidence and decentralized consensus, ensuring that knowledge updates are transparent and cryptographically secured.

**Federated and blockchain-based learning.** There is growing interest in combining federated learning (FL) with blockchain to secure distributed model training. In blockchain-based federated learning (BFL), participants collaboratively train models while blockchain nodes validate and record updates [9]. Such systems address the single-point-of-failure of central servers and can automatically flag malicious participants via smart contracts [9]. Our multi-chain design is inspired by these ideas: we view knowledge updates in our system as analogous to model updates in FL. The layered blockchain acts as a decentralized aggregator, preventing unauthorized changes and rewarding honest agents. However, unlike typical BFL which treats model weights as black boxes, our method stores explicit logical facts and proofs. This makes the learning process fully explainable and auditable, rather than a hidden model update.

**AGI testbeds and multi-agent systems.** Validating general intelligence remains challenging, and emerging testbeds focus on rich, open-ended environments. Benchmarks like the Abstraction and Reasoning Corpus (ARC) emphasize flexibility and transfer, while multi-agent simulations (e.g. OpenAI's hide-and-seek, Minos, Habitat) stress general problem solving and collaboration. Beyond game-based tests, our metaverse evaluation incorporates socio-cognitive elements (e.g. negotiating norms, legal reasoning) inspired by works on AI agents in social contexts [7]. For instance, Chen et al. introduce a "Proof-of-Thought" consensus in multi-agent systems to reward meaningful collaboration [8]. Our testbed similarly rates agent proposals using on-chain validation and reputation mechanisms, aligning with these multi-agent research themes. In summary, while prior work has explored individual aspects—neural–symbolic reasoning [4, 9], blockchain-enabled learning, or federated AI—the present framework is unique in unifying all these strands into a single, scalable AGI architecture. This integration allows us to leverage the strengths of each field simultaneously, advancing beyond state-of-the-art hybrid systems and decentralized AI platforms.

## 3 The task-based Methodology: LLMs, Logic, and Multi-Blockchain Systems

To overcome the intrinsic limitations of current large language models (LLMs) in reasoning and continuous learning, we propose a task-based methodology that integrates three key components: pre-trained LLMs, symbolic logic engines, and a hierarchical multi-blockchain knowledge store. LLMs excel at pattern recognition and rapid generation, but as recent analyses confirm, they cannot learn all computable functions and will inevitably hallucinate outside of their training data. Moreover, LLMs are typically trained on static datasets and suffer catastrophic forgetting when adapted to new domains. By contrast, symbolic logic provides rigorous inference and explicit knowledge representation.

The multi-blockchain architecture serves as a decentralized knowledge base and audit trail, enabling ongoing accumulation and verification of facts. Together, these components form a hybrid cognitive system: the LLMs provide fast, flexible reasoning, the logic modules ensure correctness and compositionality, and the blockchains enforce trust, versioning, and continuous knowledge updates.

**LLM Modules:** High-capacity neural models (e.g. transformers) act as fast heuristic subsystems, generating fluent hypotheses and handling perception-to-text tasks. They process raw input (e.g. natural language questions or sensor observations) to produce candidate solutions.

**Logic Engines:** Symbolic reasoners (e.g. theorem provers, constraint solvers) act as slow deliberative subsystems. They consume structured problem representations and database queries to perform high-precision inference, compliance checking, and formal planning.

**Multi-Blockchain Layer:** A tiered blockchain network stores training corpora, factual triples, learned rules, and world-state updates. Lower-level (private or permissioned) chains record high-throughput data and intermediate inferences, while upper-level (public) chains consolidate hashed summaries for global integrity. Smart contracts deployed on each chain implement logic rules and orchestrate interactions between LLM outputs and symbolic knowledge.

### 3.1 Logic-Based Slow Systems

Tasks requiring verifiable reasoning (e.g., theorem proving) use probabilistic first-order logic. Let a probabilistic knowledge unit be a triple:

$$< F(x, y), y = t(x), p >  \tag{1}$$

where $F(x, y)$ is a logical formula describing the task, $y = t(x)$ is a solution, and $p$ is the probability. Knowledge units are ordered hierarchically:

$$< F_1, t_1, p_1 > \preceq < F_2, t_2, p_2 > \Leftrightarrow F_1 \subseteq F_2 \& p_1 \leq p_2  \tag{2}$$

### 3.1.1 Facts and Probabilistic Knowledge Generation

A fact is formally defined as a tuple:

$$< F(c_1, c_2) >, c_2 = t(c_1) >  \tag{3}$$

where $F(c_1, c_2)$ represents a logical formula derived from probabilistic knowledge after grounding variables $x \to c_1$ and $y \to c_2$, and $t(c1)$ denotes the solution to the task $y = t(x)$. This structure encodes verifiable truths extracted from the knowledge hierarchy.

Some details:

- Facts are often derived from probabilistic knowledge triples $< F(x, y), y = t(x), p >$ by instantiating $x$ and validating $y = t(x)$ against the blockchain-stored knowledge graph.

- New probabilistic knowledge can emerge inductively from facts.

### 3.1.2 Probabilistic Logical Inference

The system leverages hierarchical probabilistic knowledge to perform inference akin to modus ponens:

$$A, A \rightarrow B \vdash B$$

We can use the same rule, but with some limitations, for probabilistic truth.

This mechanism ensures:

- Monotonic Reasoning: Higher-tier knowledge (with greater p) takes precedence.
- Uncertainty Propagation: Probabilities decay across inference chains, requiring validation against blockchain-stored facts

## 3.2 Integration with Metaverse and Blockchain

- **Fact Logging:** All validated conclusions are stored on Layer 1 3.3.5 blockchain for auditability.
- Agent Interaction: In the metaverse 5, agents use facts to enhance their cognitive abilities and test logical reasoning.

This expansion aligns with the paper's theoretical framework (e.g., Theorem on unbounded growth) and implementation details (e.g., multi-blockchain architecture). It emphasizes the bidirectional flow between probabilistic knowledge and facts, ensuring rigorous validation while enabling adaptive learning.

## 3.3 Multi-Blockchain Architecture

### 3.3.1 Inductive Definition of a Multi-Blockchain

A multi-blockchain is recursively defined as follows:

1. **Base Case**: If $B$ is a blockchain, then $\langle B \rangle$ is a multi-blockchain.
2. **Inductive Step**: If $M_1, M_2, \ldots, M_k$ are multi-blockchains and $B$ is a blockchain, then $\langle B, \langle M_1, M_2, \ldots, M_k \rangle \rangle$ is a multi-blockchain.

### 3.3.2 Height and Level Definitions

1. **Base Case**: For $\langle B \rangle$:
$$\text{level}(B) = 1, \quad h(\langle B \rangle) = 1.$$

2. **Inductive Step**: For $\langle B, \langle M_1, \ldots, M_k \rangle \rangle = M^*$:
$$h(M^*) = \text{level}(B) = \max\{h(M_1), \ldots, h(M_k)\} + 1.$$

### 3.3.3 Master Blockchain

The **master blockchain** of a multi-blockchain $M$ is the blockchain $B$ within $M$ such that:
$$\text{level}(B) = h(M).$$

### 3.3.4 Predicate Multi

The predicate $\text{Multi}(X)$ holds if and only if $X$ is a valid multi-blockchain under the inductive rules above.

**Lemma** (Hierarchical Consistency) If $h(M) > 1$, then for all $x \in \text{head}(M)$:

$$Multi(X) \ \wedge \ h(x) \in [1, h(M) - 1] \tag{4}$$

**Proof:** By induction: the components $M_1, \ldots, M_k$ in $\langle B, \langle M_1, \ldots, M_k \rangle \rangle$ must themselves be multi-blockchains of strictly lesser height.

### 3.3.5 A special case of multi-blockchain

A layered blockchain stores knowledge:

- **Layer 1 (Fast):** High-throughput chains (e.g., Solana) store raw training data.
- **Layer 2 (Intermediate):** Chains like Ethereum execute smart contracts for knowledge aggregation.
- **Layer 3 (Master):** A decentralized, immutable chain (e.g., Bitcoin-like) stores hashed knowledge digests.

Data flows upward via periodic Merkle root commitments:

$$Commit(L_i) = H(Data_{L_i}||sign) \rightarrow L_{i+1} \tag{5}$$

## 4 LLM's Interaction with the Knowledge Base

### 4.1 Smart Contracts and Logic Module Coordination

In our framework, smart contracts are self-contained programs stored and executed within a single blockchain. While a smart contract typically cannot access the entire multi-blockchain structure directly, this limitation is resolved through a dedicated logic module—an external subsystem that monitors the entire multi-blockchain in real-time. This logic module acts as a global orchestrator, capable of:

1. Querying data across all blockchain layers.
2. Triggering specific smart contracts with tailored parameters.
3. Mediating interactions between smart contracts, LLMs, and the knowledge hierarchy.

### 4.2 Task-Based Problem-Solving Workflow

The methodology revolves around a structured problem-solving pipeline:

**1. Initial Knowledge Check:**

- When an AI agent encounters a task, the logic module first scans the multi-blockchain's knowledge base for pre-validated solutions.
- Example: A query like "Diagnose reactor overheating" triggers a search for verified causal relationships (e.g., <F: "Low coolant flow $\Rightarrow$ Overheat", t: "PumpA", p=0.93>).

**2. LLM Engagement:**

- If no solution exists, the logic module forwards the task to the LLM, augmented with probabilistic knowledge units from the blockchain.
- The LLM generates hypotheses and reasoning chains (e.g., "Pump failure due to corrosion (Steps: 1. Sensor data shows RPM drop; 2. Maintenance logs indicate rust)").

**3. Smart Contract Validation**

- The logic module invokes domain-specific smart contracts to verify the LLM's solution.
- Example: A $ThermodynamicCompliance$ smart contract checks whether the hypothesis aligns with physical laws stored on Layer 2.

**Gap Analysis and Iteration:**

- Invalid solutions trigger the $LogicEngine.get\_proof\_gaps$ method, which identifies missing premises (e.g., "Missing corrosion data for PumpA in 2024").
- The logic module queries Layer 1 for raw sensor logs, updates the context, and iterates until confidence thresholds are met.

256 **More detailed:** LLM is trying to solve this problem based on the data that is stored at different levels
257 of the multi-blockchain network, taking into account its hierarchical structure.

258 In the context of this framework, a hypothesis refers to a candidate solution or proposed answer
259 generated by the LLM in response to a query. It is a tentative claim that must be rigorously validated
260 against the blockchain-stored knowledge base and logic rules before being accepted as reliable.

---

**Algorithm 1** class Hypothesis

---

```
1: class Hypothesis:
2:   def __init__(self, content: str, reasoning: str, confidence: float):
3:     self.content = content # Claim ("Valve VX-2 failed")
4:     self.reasoning = reasoning # Proof steps ("1. Corrosion detected...")
5:     self.confidence = confidence # 0-1 validation score
6:     self.proof = None # Formal proof object from LogicEngine
```

---

261 The $LogicEngine.validate$ function in Algorithm 2 checks a hypothesis and its reasoning steps
262 against existing knowledge stored in the blockchain. If there are gaps or missing information that
263 prevent the hypothesis from being fully validated, '$LogicEngine.get\_proof\_gaps$' should identify
264 these gaps.

265 For example, missing data (like sensor logs), unverified logical steps (like an unsupported lemma in
266 a proof), or contradictions with existing knowledge. The function needs to analyze the validation
267 results and the proof steps provided by the LLM to pinpoint where the reasoning falls short.

268 The pseudocode mentions examples like 'Missing lemma about X'. This suggests that the feedback
269 needs to be specific enough to guide the next query to the blockchain. The function might categorize
270 gaps into different types (missing data, logical inconsistencies, incomplete proofs) and generate
271 structured feedback for each type.

272 Another aspect is how the gaps are prioritized. If multiple gaps are identified, which one should be
273 addressed first? The function might prioritize gaps based on their impact on the overall confidence
274 score or the hierarchy of the knowledge base.

275 Finally, the $LogicEngine.get\_proof\_gaps$ method analyzes failed hypothesis validations to identify
276 precise missing components in the reasoning chain. It generates structured feedback to guide iterative
277 knowledge retrieval from the blockchain, enabling the system to resolve ambiguities or incomplete
278 proofs.

279 This integration of LLMs, logic modules, and multi-blockchains establishes a robust foundation for
280 trustworthy, scalable AGI systems capable of tackling open-world challenges.

## 281 5   The Role of the Metaverse and Multi-Agent Systems

282 The metaverse is envisioned as a persistent, shared virtual world that integrates advanced networking,
283 virtual/augmented reality, and AI. In our context, it serves as a rich testbed for AGI: an open-ended
284 multi-agent environment where physical and virtual entities co-exist and interact. Formally, we
285 treat the metaverse as a multi-agent system (MAS) [13] $\mu$ with N agent "slots" , each receiving
286 observations and taking actions in a common environment.

287 Specifically, $\mu$ can be modeled as a tuple $(S, \{A_i\}, T, \{R_i\}, \{O_i\})$ where $S$ is the (possibly shared)
288 state space, $A_i$ is the action space of agent $i$, $T$ defines transition probabilities, $R_i$ is the reward (or
289 objective) function, and $O_i$ specifies observations of each agent. Each agent interacts in discrete time
290 steps: at step $t$, agent $i$ receives observation $o_{i,t} \in O_i$, takes action $a_{i,t} \in A_i$, and obtains reward
291 $r_{i,t \in R_i}$, with the join profile $(a_{1,t}, \ldots, a_{N,t})$ driving the environment to the next state.

292 In such a social setting, agents may form teams, compete, or cooperate, emulating the complexity
293 of human society. The general view of the interaction between an agent and an environment can be
294 extended to multiple agents by letting them interact simultaneously with the environment. Indeed, a
295 multi-agent metaverse can exhibit emergent phenomena (e.g. markets, cultures, conflict) inaccessible
296 to isolated agents.Within this metaverse, diverse agents interact with the logic and blockchain
297 infrastructure. We distinguish three broad agent classes:

**Virtual AI agents:** Purely synthetic entities (software bots) endowed with the dual-system cognitive architecture described above. These agents communicate, form beliefs, and act under the same logic-and-LLM paradigm. They can post new knowledge to the blockchain, propose logical rules via smart contracts, or collaborate on tasks like exploration.

**Cyber-physical agents:** Robotic or IoT systems that bridge the virtual and physical. For example, a delivery robot in the real world whose decisions are partly managed by its digital avatar in the metaverse. Such agents use sensory inputs (possibly augmented by VR simulators) and push real-world data into the virtual ledger, allowing digital reasoning about physical events.

**Biological (human) agents:** Human participants, embodied as avatars or digital twins [10], engage in the metaverse. They can query knowledge via natural language (LLM) or refer to formal rules (through the logic interface). Their actions—whether trading digital assets or voting on policies—are recorded on the blockchain, and they experience the consequences of the shared virtual economy.

All agents access the global knowledge fabric by interfacing with logic modules and the blockchain. Logic modules serve as query engines for rule-based reasoning. For instance, an agent might pose a question via a smart contract that triggers a logical proof search or ethical check (e.g. "Is this action permitted under the current laws?"). The multi-blockchain network captures every significant event: transactions, contracts, sensory logs, even agent dialogues. Importantly, a blockchain can enforce consensus on shared rules. For example, an autonomous "legislative DAO" agent could submit a proposed regulation (encoded symbolically) to a public chain; other agents (or humans) vote via transactions, and the rule is enacted if consensus is reached. This creates an evolving digital constitution.

Such a metaverse ecosystem is invaluable for studying AGI safety, reasoning, and ethics at scale. First, safety testing can occur in a simulated environment where consequences are contained. We can deliberately introduce adversarial scenarios or ethical dilemmas without real-world risk. For example, autonomous legal agents might rule on counterfactual cases to test alignment. Second, the complexity of the environment stresses the AGI's reasoning abilities: multi-step social games, unpredictable agent behaviors, and large state spaces reveal how well the AI generalizes and maintains coherence. Third, ethical norms can be encoded in the logic modules and enforced via the ledger (e.g. immutable human rights protocols), allowing rigorous evaluation of value alignment. The transparency of blockchains ensures that any undesirable behavior is traceable. Fourth, the metaverse allows massive scaling: thousands of agents can coexist, enabling studies of population dynamics, emergent cooperation/competition, and distributed consensus. Prior work emphasizes that MAS can solve problems impossible for single agents, and our platform embodies this by hosting both cooperative teams and adversarial groups.

Finally, consider practical applications. In a virtual legal system, autonomous judges (AI agents) use logic contracts to interpret a shared code of laws, while human attorneys submit evidence as blockchain transactions. Disputes are resolved by the consensus of validator nodes (simulating juries), and all verdicts are logged on an immutable blockchain ledger for accountability. In scientific research, teams of AI and human scientists inhabit a simulated laboratory. Agents propose experiments using LLMs (for hypothesis generation) and formal models (for theoretical analysis); results are automatically recorded on the blockchain, enabling reproducibility and automated meta-analysis by logic agents. In adaptive education, AI tutor agents use LLMs to generate personalized lessons but check pedagogical rules via logic modules; student progress is tracked on-chain so that both human teachers and AI can assess learning outcomes. In each case, the metaverse acts as a comprehensive sandbox: it harnesses VR/AR for embodied interaction, blockchains for secure data provenance, and hybrid AI for intelligent agency. This combination provides a rich testbed where strong AI systems can be rigorously evaluated on safety, reasoning, and ethical behavior before any real-world deployment.

The framework builds on recent advances in neural-symbolic AI, federated LLM training with blockchains, and metaverse-AI integration, among others. All design choices are motivated by the goal of creating an AGI system that is robust, interpretable, and aligned within an interactive multi-agent world.

## 6  Theoretical Guarantees: Unbounded Cognitive Growth

**Theorem** (Asymptotic Cognitive Scaling) Assuming infinite computational resources and a monotonically increasing knowledge hierarchy, the system's problem-solving capability grows without bound.

Proof Sketch:

Let $K_t$ be the knowledge set at time $t$, and $P(K_t)$ be the set of solvable problems.

When axiomatizing a theory, we effectively define the set of provable theorems within that theory. By deriving logical consequences from existing axioms and theorems, we can iteratively expand the set of provable statements.

Formally, let $T$ be a theory with $Axioms$ as its set of axioms. The set of theorems $Theorems$ can be defined inductively as the least fixed point of the monotone operator $\Gamma$, where:

$$Q^* = \Gamma(Q) = \{q \in T \mid \exists q_1, \ldots, q_n \in Q : q_1, \ldots, q_n \vdash q\} \tag{6}$$

This operator $\Gamma$ captures the closure of $Q$ under logical entailment, ensuring that all derivable statements are included in $Q^*$.

We have that on each iteration: $P(K_t) \subset P(K_{t+1})$. As $t \to \infty$, $lim_{t \to \infty} |P(K_t)| = \infty$.

## 7  Ethical Considerations

- **Autonomy vs. Control:** Agents may self-modify smart contracts, risking unintended goals. We propose ethical governor modules that override harmful actions.
- **Decentralization:** Master blockchain consensus prevents single-entity control.

## 8  Conclusion

We have presented a novel hybrid framework for Strong AI that synergizes LLM-based inference, symbolic logic reasoning, and a decentralized blockchain memory. By structuring knowledge into a hierarchical probabilistic logic and storing it across tiered blockchains, our system achieves continuous, auditable learning: agents can ingest new information on-chain, verify it rigorously, and reliably expand their capabilities over time. This design overcomes classic LLM weaknesses (hallucinations, outdated facts) by always checking outputs against formally verifiable knowledge. Using the example of a metaverse built according to our methodology, we have shown how agents can interact with each other and how their cognitive abilities can grow in solving problems with a fixed budget of computing resources, which underlines the potential of the approach. Key outcomes include (1) formal guarantees of asymptotic cognitive growth with the knowledge hierarchy model, (2) practical deployments in domains like law, science, and education enabled by hybrid AI–blockchain coordination, and (3) built-in ethical safeguards (e.g. on-chain constitutional rules and universal basic income mechanisms) that align agent behavior with human values.

Looking forward, our roadmap envisions several extensions and broader impacts. First, we aim to enhance the blockchain substrate: adopting quantum-resistant consensus schemes and optimizing cross-chain interoperability will enable truly global knowledge networks. Second, we will integrate richer ontologies and continuous learning protocols, perhaps informed by federated learning advances, to accommodate real-world data heterogeneity and privacy. Third, interdisciplinary collaboration will be crucial: working with cognitive scientists, legal scholars, and ethicists, we plan to refine the logical knowledge representation and ethical frameworks embedded in smart contracts. Over the long term, we anticipate a worldwide ecosystem of intelligent agents sharing verifiable knowledge and solving complex societal challenges in tandem. Such a system could revolutionize fields from law (automated treaty analysis) to science (collaborative discovery) by guaranteeing that AI decisions are both powerful and transparent. Ultimately, by converging machine learning, formal reasoning, and secure decentralized technology, our task-based methodology lays the groundwork for robust, trustworthy AGI. We hope this work will inspire further research on hybrid architectures and usher in a new era where AI systems learn and evolve in a provable, collaborative manner—truly embodying the vision of Strong AI.

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

# A   Algorithm – Verify Compliance

The Verify Compliance algorithm ensures that hypotheses generated by LLMs adhere to the knowledge stored in the multi-blockchain system. It iteratively refines solutions by resolving gaps in reasoning through cross-layer blockchain queries and logic-driven validation. Below is a detailed breakdown of its components and workflow.

**Workflow Steps**

**1. Initialization:**

- Fetch context from the blockchain using the query and $blockchain\_id$.
- Initialize variables to track the best solution and iteration count.

**2. Hypothesis Generation:**

- The LLM generates candidate hypotheses with structured reasoning chains (e.g., "Hypothesis|Step 1; Step 2; ...").

**3. Validation Loop:**

- For each hypothesis:
  - The LogicEngine validates both the hypothesis and its reasoning steps against blockchain-stored knowledge.
  - A confidence score (0–1) is assigned based on alignment with verified facts.
- Early exit if any hypothesis exceeds the confidence threshold.

**4. Context Refinement:**

- If validation fails, the $LogicEngine$ identifies missing premises (e.g., unverified lemmas or sensor data gaps).
- Supplementary data is queried from lower blockchain layers (e.g., raw sensor logs from Layer 1).

**5. Termination:**

- After $max\_iterations$, log unresolved queries with partial proofs and missing steps

**Algorithm 2** Verify compliance

```
 1: function SOLVE_QUERY(query, blockchain_id, max_iterations=5, confidence_threshold=0.9)
 2:     # Initial knowledge fetch
 3:     current_context = blockchain.query(blockchain_id, query)
 4:     best_solution = None
 5:     iteration = 0
 6:     while iteration < max_iterations:
 7:     # Generate hypotheses with reasoning chains
 8:     hypotheses = LLM.generate(
 9:       prompt=query,
10:       context=current_context,
11:       response_format="hypothesis|reasoning" # Structured output
12:     )
13:     # Parse and validate hypotheses with their reasoning
14:     validated = []
15:     for raw_hyp in hypotheses do
16:         hyp_text, reasoning = raw_hyp.split("|", 1) # Split into components
17:         hyp = Hypothesis(
18:           content=hyp_text,
19:           reasoning=reasoning,
20:           confidence=0.0
21:         )
22:         # Validate both hypothesis and its reasoning chain
23:         proof, hyp.confidence = LogicEngine.validate(
24:           hypothesis=hyp.content,
25:           proof_steps=hyp.reasoning,
26:           blockchain_id=blockchain_id,
27:           context=current_context
28:         )
29:         hyp.proof = proof # Store formalized proof object
30:         validated.append(hyp)
31:         # Early return if high-confidence solution found
32:         if hyp.confidence > confidence_threshold:
33:           blockchain.commit(
34:             blockchain_id=blockchain_id,
35:             data="hypothesis": hyp.content, "proof": hyp.proof,
36:             contract="KnowledgeUpdate"
37:           )
38:           return hyp
39:         # Update best solution using combined confidence/proof metrics
40:         current_best = max(validated, key=lambda x: x.confidence, default=None)
41:         if current_best and (best_solution is None
42:             or current_best.confidence > best_solution.confidence):
43:           best_solution = current_best # Refine context using proof failures
44:         feedback = LogicEngine.get_proof_gaps(validated) # e.g., "Missing lemma about X"
45:         supplementary_data = blockchain.query(
46:           blockchain_id=blockchain_id,
47:           query=feedback,
48:           depth=iteration+1,
49:         proof_aware=True # Prioritize proof-related knowledge
50:         )
51:         current_context += supplementary_data
52:         iteration += 1
53:     # Fallback with proof-aware logging
54:     if best_solution:
55:       blockchain.log(
56:         query,
57:         status="PartialProof",
58:         missing_steps=LogicEngine.get_proof_gaps([best_solution])
59:       )
60:       return best_solution
61:     return "No solution with valid proof found"
```