# OpenReview forum: "The Task-Based Methodology of Strong AI: Integrating LLMs, Logic Reasoning, and Multi-Blockchain Architectures"
_NeurIPS.cc/2025/Position_Paper_Track — Submitted to NeurIPS 2025 Position Paper Track_

### Official Review · Reviewer_xku9 · 2025-08-04

**Significance:** 1
**Presentation:** 1
**Rating:** 1
**Confidence:** 4

**Summary:**

The paper describes a neurosymbolic system which combines an LLM, with a symbolic engine and blockchain based memory. These new additions are supposed to address some of the current failings of LLMs such as hallucinations, outdated information and complex reasoning. The symbolic engine is supposed to first decide if there is an existing solution and if not the LLM is activated with knowledge fro the blockchain. LLMs then generate hypothesis which are verified through smart contracts. The paper also references the metaverse as a way to test this new system.

**Strengths:**

The paper discusses some of the current issues with LLMs. Decentralized memory could be an interesting idea depending on the details.

**Weaknesses:**

- The paper has no position. The introduction sounds like a main-track submission.
- The system the paper describes is extremely vague. The definitions do not make sense and have little connection to real world use cases. In general the paper seems to use 'buzz words' without any real explanation.  For example, the paper assumes that symbolic systems are able to  "consume structured problem representations and database queries to perform, high-precision inference, compliance checking, and formal planning" (L161)". How are such systems supposed to generalize?  What is the blockchain described? How is it supposed to perform all the functions described?
- Entire metaverse discussion is extremely unclear.  Not only is the 'metaverse' undefined, it seems to be totally unrelated to the system described. The discussion of using it as a 'testbed' seems thoroughly unrelated given that no such thing actually exists. On top of that, there are a multitude of problems with the 'real world applications.' For example, it assumes that things such as laws can be algorithmically determined. In many cases, judges rule when the law is ambiguous i.e. there is not algorithmic decision.
- The paper is unorganized and not well written.

**Questions:**

- What is the position?
- How is such a system supposed to be built?
- What is the blockchain definition used here? How is it supposed to be built?
- How are symbolic systems supposed to generalize?
- What is the metaverse?
- What are the related works this is based on?

**Alternative Position:**

No

**Author Identification:**

No.

**Context:**

1

**Discussion:**

1

**Ethics:**

["NO or VERY MINOR ethics concerns only"]

**Position:**

No, the paper presents new research without clearly advocating a position.

**Support:**

1

**Thoroughness:**

3

---

### Official Review · Reviewer_zLor · 2025-08-09

**Significance:** 1
**Presentation:** 2
**Rating:** 1
**Confidence:** 4

**Summary:**

The paper advocates a *task‑based hybrid methodology* by tightly coupling (i) LLM‑driven hypothesis generation, (ii) probabilistic symbolic‑logic verification, and (iii) a hierarchical multi‑blockchain memory. The authors claim this triad mitigates LLM hallucinations, enables explainable reasoning, and supports lifelong, auditable knowledge growth. They outline a layered blockchain design, provide formal definitions (e.g., inductive multi‑blockchain, probabilistic knowledge hierarchy), and offer a proof sketch of *asymptotic cognitive scaling*. A simulated “metaverse” populated by heterogeneous agents is proposed as an evaluation test‑bed, but empirical results remain largely conceptual.

**Strengths:**

1. Ambitious unification of neurosymbolic reasoning and decentralized provenance
2. No significant format errors

**Weaknesses:**

1. **Lack of views.** No position is stated explicitly. No alternative views for sure.
2. **Empirical vacuum.** No quantitative evaluation accompanies the metaverse test‑bed, claims of “unbounded cognitive growth”, etc.
3. **Insufficient contexts.** Too few contexts are provided which makes me even unsure about the domain of this work.

**Questions:**

**Unclear purpose of writing:** What is the main point to be conveyed in the paper?

**Alternative Position:**

No

**Author Identification:**

No.

**Context:**

1

**Discussion:**

1

**Ethics:**

["NO or VERY MINOR ethics concerns only"]

**Position:**

No, the paper argues that a specific technical approach is superior to other approaches.

**Support:**

1

**Thoroughness:**

2

---

### Official Review · Reviewer_LEqi · 2025-08-12

**Significance:** 3
**Presentation:** 3
**Rating:** 5
**Confidence:** 4

**Summary:**

The paper presents an three architectural vision for achieving Artificial General Intelligence (AGI), built on the integration of three distinct technological pillars. The first is a a cognitive engine that combines Large Language Models (LLMs) for rapid, creative hypothesis generation (“System 1”) with symbolic logic engines for rigorous, auditable verification (“System 2”). The second is a decentralized knowledge base implemented through a hierarchical multi-blockchain architecture, designed to provide an immutable, universally trusted, and continuously evolving memory for the AGI. The third pillar is a large-scale, persistent metaverse populated by AI, cyber-physical, and human agents, serving as a dynamic testbed to evaluate the AGI’s cognitive development, emergent behaviors, and safety alignment. The core argument is that this synthesis can address the key shortcomings of current AI systems—such as hallucinations, static knowledge, and limited trustworthiness—while enabling a scalable, interpretable, and ethically aligned path toward AGI.

**Strengths:**

The paper’s primary strength lies in its ambitious and thought-provoking vision, bringing together neuro-symbolic AI, decentralized ledgers, and multi-agent simulations into a unified architecture for AGI. It builds a compelling case by framing the work around pressing, high-impact challenges in the field, including LLM hallucination, knowledge verification, and AI safety. The topic is highly relevant to the NeurIPS community, as it addresses the grand challenge of achieving AGI. its call for a rich, dynamic metaverse testbed recognizes the limitations of static benchmarks and the need for more adaptive, interactive evaluation methods for advanced AI.

**Weaknesses:**

The paper's central argument is critically weakened by its failure to address the Blockchain tri-lemma problem. It proposes a real-time, high-throughput knowledge base on a technology that is fundamentally constrained in scalability, making the core architecture infeasible without addressing this conflict.

The paper also significantly downplays the "autoformalization" challenge—the unsolved problem of reliably converting natural language into formal logic.  An alternative position not considered is a hybrid knowledge store: using a high-performance off-chain database (like a vector DB) for real-time operations, while using the blockchain periodically to store immutable hashes for auditing pupose. This would achieve the goal of verifiability without sacrificing the performance required for a cognitive loop. The reliance on a technologically immature metaverse concept for evaluation also adds a layer of impracticality.

**Questions:**

My questions majorly stem from the weaknesses that I have mentioned above, as I feel the authors can clarify some of these questions and can start a constructive discussion that questions the proposed methods in a healthy way.

1. The paper's scaling theorem assumes "infinite computational resources."  How do you reconcile this theoretical guarantee with the practical and often prohibitive computational and financial costsof the proposed blockchain implementation?

2. What are the specific advantages of using a full-scale metaverse for evaluation over more controlled, targeted multi-agent simulations, which have proven effective at studying emergent behavior without the immense technical overhead and potential confounding variables?

3. Your framework's real-time cognitive loop requires (rather high) frequent interaction with the knowledge base. How do you reconcile this with the inherent latency and low throughput of blockchain consensus(Blockchain Tri-lemma)?

**Alternative Position:**

Yes, and alternative positions are well-considered and named but not addressed

**Author Identification:**

No.

**Context:**

2

**Discussion:**

4

**Ethics:**

["NO or VERY MINOR ethics concerns only"]

**Position:**

Yes, the paper argues for or against a position related to machine learning.

**Support:**

2

**Thoroughness:**

4

---

### Meta-Review · Area_Chair_LFCP · 2025-09-12

**Rating:** 4
**Confidence:** 4

**Strengths:**

The paper attempts an ambitious and provocative vision for AGI, integrating three pillars: neuro-symbolic reasoning (LLMs + logic engines), decentralized blockchain-based memory, and a metaverse-style multi-agent testbed.

**Weaknesses:**

The central reliance on blockchain for a high-throughput, real-time knowledge base appears technically infeasible due to scalability. The paper downplays unresolved challenges such as autoformalization. The metaverse proposal is vague. Alternative approaches (e.g., hybrid off-chain databases, more lightweight multi-agent simulations) are not substantively considered.

**Questions:**

How do the authors reconcile the performance and cost limitations of blockchain consensus with the proposed real-time cognitive loop?
Why is a full metaverse required for evaluation, rather than more targeted and feasible multi-agent simulations?

**Ethics:**

There are no significant ethical concerns.

**Thoroughness:**

4

---

### Decision · Program_Chairs · 2025-09-26

Reject